# Bronchoalveolar Lavage and Blood Markers of Infection in Critically Ill Patients—A Single Center Registry Study

**DOI:** 10.3390/jcm10030486

**Published:** 2021-01-29

**Authors:** Jarno F. Kronberger, Thomas C. Köhler, Corinna N. Lang, Markus Jäckel, Xavier Bemtgen, Tobias Wengenmayer, Alexander Supady, Wolfram Meschede, Christoph Bode, Viviane Zotzmann, Dawid L. Staudacher

**Affiliations:** 1Department of Cardiology and Angiology I, Heart Center, Faculty of Medicine, University of Freiburg, 79106 Freiburg, Germany; jarno.Kronberger@uniklinik-freiburg.de (J.F.K.); corinna.lang@uniklinik-freiburg.de (C.N.L.); markus.jaeckel@uniklinik-freiburg.de (M.J.); xavier.bemtgen@uniklinik-freiburg.de (X.B.); tobias.wengenmayer@uniklinik-freiburg.de (T.W.); Alexander.supady@uniklinik-freiburg.de (A.S.); christoph.bode@uniklinik-freiburg.de (C.B.); viviane.zotzmann@uniklinik-freiburg.de (V.Z.); 2Department of Internal Medicine III, Medical Intensive Care, Medical Center, University of Freiburg, 79106 Freiburg, Germany; 3Department of Pneumology, Faculty of Medicine, University of Freiburg, 79106 Freiburg, Germany; thomas.koehler@uniklinik-freiburg.de (T.C.K.); wolfram.meschede@uniklinik-freiburg.de (W.M.)

**Keywords:** bronchoalveolar lavage (BAL), microbiological testing, C-reactive protein (CRP), procalcitonin (PCT), immunocompromised, intensive care unit (ICU)

## Abstract

Microbiological sampling is an indispensable targeted antibiotic therapy for critically ill patients. Invasive respiratory sampling by bronchoalveolar lavage (BAL) can be performed to obtain samples from the lower respiratory tract. It is debated as to whether blood markers of infection can predict the outcome of BAL in a medical intensive care unit (ICU). Retrospectively, all ICU patients undergoing BAL from 2009–2018 were included. A total of 468 BAL samples from 276 patients (average age 60 years, SAPS2 47, ICU-mortality 41.7%) were analyzed. At the time of BAL, 94.4% patients were mechanically ventilated, 92.9% had suspected pneumonia, 96.2% were on antibiotic therapy and 36.3% were immunocompromised. Relevant bacteria were cultured in 114/468 (24.4%) cases of BAL. Patients with relevant bacteria in the culture had a higher ICU mortality rate (45.6 vs. 40.4%, *p* = 0.33) and were significantly less likely to be on a steroid (36 vs. 52%, *p* < 0.01) or antimycotic (14.9 vs. 34.2%, *p* < 0.01), while procalcitonin (PCT), C-reactive protein (CRP), and white blood cell (WBC) counts were similar. The area under the receiver operating curve (AUC) values for positive culture and PCT, CRP and WBC counts were low (0.53, 0.54 and 0.51, respectively). In immunocompromised patients, AUC values were higher (0.65, 0.57 and 0.61, respectively). Therefore, microbiological cultures by BAL revealed relevant bacteria in 24.4% of samples. Our data, therefore, might suggest that indication for BAL should not be based on blood markers of infection.

## 1. Introduction

In critically ill patients on intensive care units, hospital-acquired pneumonia (HAP) is a common complication, especially for those undergoing mechanical ventilation (ventilator-associated pneumonia, VAP). Data suggest that the incidence rate of VAP is as high as 16–18 cases per 1000 ventilator days with an attributable mortality rate of 6–8% [1,2]. Current guidelines recognize this considerable health issue and recommend that antibiotic therapy is given for hospital-acquired pneumonia after microbiological sampling [3,4]. Invasive respiratory sampling methods like bronchoalveolar lavage (BAL) are conducted for microbiological sampling; this method is performed in roughly 18% of all patients in European intensive care units (ICUs) [2]. Since invasive respiratory sampling might endanger critically ill patients (e.g., worsening of oxygenation) [2,5,6], especially when performed by an inexperienced investigator, noninvasive sampling is a reasonable alternative for the diagnosis of pneumonia [3,7]. Besides positive microbiological samples, pneumonia frequently presents with elevated markers of infection including C-reactive protein (CRP), an elevated or lowered white blood cell (WBC) count, and procalcitonin (PCT) [8,9,10]. While it is recommended to base the decision to start an empiric antibiotic therapy on clinical criteria alone, rather than on a combination of clinical criteria and markers of infection [3], it is unclear as to whether these markers of infection can be used to prompt invasive respiratory sampling. We, therefore, retrospectively analyzed all patients in our ICU undergoing invasive respiratory sampling and correlated the obtained microbiological results with known markers of infection.

## 2. Materials and Methods

We retrospectively analyzed the medical records of all patients treated at the medical ICU of the University Hospital, Freiburg between January 2009 and November 2018. Inclusion criteria were the occurrence of bronchoalveolar lavage (BAL) with microbiological testing. Cases of BAL not performed by a senior in respiratory care were excluded, as were cases of BAL with incomplete documentation. Patient identity was blinded in the analysis, and ethical approval was obtained (Ethics Committee of Albert-Ludwigs University of Freiburg, file number 337/18).

Clinical management: Our medical ICU is located at a university hospital that acts as a reference center for acute respiratory distress syndrome (ARDS) and extra corporeal membrane oxygenation (ECMO). In cases of suspected bacterial pneumonia, our local standard operating procedures advocate for the collection of at least two pairs of blood samples and tracheal fluid before starting empiric antibiotic therapy. According to local policy, BAL is strongly advocated in all patients with unclear severe pulmonary failure or ARDS and presumed pulmonary infection. BAL is performed by a specialist in pulmonary care who is available during working hours. The restriction to respiratory specialists has a longstanding history at our institution and has been established to guarantee maximum quality and safety for our patients. After the lavage fluid had been recovered, it was processed by the microbiological institute of the University of Freiburg. Routine diagnostics were performed for each sample, which included the creation of microbiological cultures on different standard media (blood agar, chocolate agar, MacConkey agar, yeast cysteine blood agar, GVPC (glycine, vancomycin, polymyxin B, and cycloheximide) -agar and Sabouraud dextrose agar). The identification of species was done by using MALDI-TOF (matrix-assisted laser desorption time-of-flight mass spectrometry). In cases of Streptococcus pneumoniae differentiation from other Streptococcus spp., this was achieved by using the optochin susceptibility test. The results of the microbiological diagnostics are documented in the electronic patient files.

Methods BAL: BAL was performed as reported previously [11] and carried out in the radiologically most affected lung lobe. In cases of diffuse infiltrates or involvement of multiple lobes, the middle lobe or the lingula was preferred for BAL. Bronchoscopes with a diameter of 8 mm were used to obtain a standardized wedge position. For exclusive microbiological sampling, 100 mL of sterile saline (0.9% NaCl) was instilled in 20 mL aliquots. After each instillation, the fluid was gently suctioned. If immunological or additional analysis was warranted, BAL was performed with up to 300 mL of sterile saline. The BAL aliquots (including the initial aliquot) were pooled and collected in a sterile jar and immediately transported to the laboratory for further analysis.

Definitions: A relevant microbiological microorganism is an organism detected in microbiological culture of a BAL with ≥10^3^ colony forming units/mL (CFU) that could potentially cause pneumonia. Pathogens typically not causing pneumonia were considered non-relevant (see Appendix A for detailed information on all microbiological samples detected). An even more rigid subgroup looking only at patients’ BAL with ≥10^4^ CFU/mL was also investigated and is provided in the Appendix A. In addition, all Candida spp. were considered contamination due to their common occurrence in samples and relatively low probability of causing pneumonia. Pneumonia was considered to be present when documented by the physician in charge, as diagnosed in the electronic patient files. The diagnosis of pneumonia was based on the current German guidelines for HAP [4,12].

Laboratory Parameters: The values of high-sensitivity C-reactive protein (CRP) and white blood cell (WBC) count closest to the BAL within 24 h were considered for this research. Since high-sensitivity procalcitonin (PCT) is not evaluated every day in clinical practice, the closest reported value within 48 h to the BAL was considered. Normal ranges were CRP: ≤5.0 mg/L, PCT: ≤0.5 ng/mL and WBC count: 4000–10,000 cells/mL. 

Antibiotics: For the detection of antibiotics used at the time of BAL, all antibiotics given within 24 h before the collection of BAL were considered. This included all antibiotics either administered by inhalation or applied systemically (intravenous or oral). When an antibiotic switch was performed within 24 h before collection of the BAL, all antibiotics given were counted. Since only the final time and date of the microbiological reports were available to us, and because preliminary results are often communicated directly via telephone, we could not evaluate whether an antibiotic switch was performed as a reaction to a preliminary microbiological report or if the switch was an educated guess (empiric) by the physicians in charge. 

Immunocompromisation: This was defined as current therapy with immunosuppressive drugs (excluding steroids) or chemotherapeutics (as treatment of current hematological malignancy or solid tumor) during the current hospital stay, metastatic solid tumors or hematologic malignancies being present, the status after organ transplant, leukopenia at the time of BAL according to documentation, infection with HIV or the presence of CF.

Data collection: A computerized search using the German Procedure Classification/OPS code for bronchoalveolar lavage 1-620.0, 1-620.3 and 1-843 was conducted. By manual review, inclusion and exclusion criteria were verified. Specifically, patients without electronic patient files, cases in which BAL was not performed by a pulmonary specialist during the daytime, patients with incomplete reports and cases of BAL performed outside the ICU were excluded. Patient data were accessed and analyzed on a tabular listing using Microsoft Excel (version 2010, Microsoft, Redmond, WA, USA). All outcome variables were evaluated by a manual search of medical and patient records.

Statistical analysis: Relevant data were integrated into standardized tables. For data analysis, SPSS (version 25, IBM Statistics, Armonk, NY, USA) and Prism (version 8, GraphPad, San Diego, CA, USA) were employed. For statistical analysis, the student’s t-test and Fisher’s exact test were used when appropriate, and a *p* value of <0.05 was considered statistically significant. The area under the receiver operating curve was calculated using Prism (version 8, GraphPad), and the Youden Index (Youden’s J) was calculated using the formula “(sensitivity + specificity)-100” and the highest value was taken as the Youden Index. Data are given as n (%), the median (range) or the number of patients (percentage of group) if not stated otherwise. Outcomes reported were not adjusted for clinical scores or risk factors for pneumonia.

## 3. Results

A total of 468 cases of BAL with complete biological testing results from 276 patients were included in the present analysis. Please see Figure 1 for details on recruiting. Regarding patient characteristics, at the time of the first BAL patients had a median age of 60 years, 72% were male and the rate of ICU mortality was 41.7%. Importantly, the mean number of BALs performed per patient was 1.8, and 96% of patients were already on antibiotic therapy when the BAL was performed. See Table 1 for details.

Microbiological samples: In a total of 220/468 (47.0%) microbiological samples obtained by BAL, a bacterial pathogen could be cultivated. Of these, 114/468 (24.4%) were considered relevant bacterial pathogens. When comparing patients with and without relevant bacteria in BAL, we found that both groups were comparable for all investigated parameters including ICU mortality (45.6 vs. 40.4%, *p* = 0.33). Patients with relevant bacteria, however, were significantly less likely to be on a steroid (36 vs. 52%, *p* < 0.01) or antimycotic (14.9 vs. 34.2%, *p* < 0.01) compared with patients without relevant bacteria. Detailed information on the microbiological cultures obtained is given in Table 2. The correlations mentioned above did not significantly change when the more rigid approach (stricter definition of relevant bacteria and only considering cultures with ≥10^4^ CFU/mL) was used. The number of relevant bacteria with this approach was 56/468 (12.0%); see Appendix A.

Prediction of relevant pathogens in BAL cases by parameters of infection: Laboratory parameters of infection (PCT, CRP and WBC count) were similar between the groups. Evaluation of the area under the receiver operating curve for the three parameters suggested that all three parameters had no predictive value for the presence of relevant pathogens, and the optimal cutoff according to the Youden index confirmed these findings; see Figure 2. Excluding patients with clinically relevant fungi in BAL or using a more rigid definition of relevant bacteria also showed a similarly low area under ROC values for PCT, CRP, and WBC counts; see Appendix A.

Immunocompromised subgroup: When evaluating the subgroup of immunocompromised patients, we found that the values for the area under the receiver operating curve for PCT, CRP and WBC counts were higher compared to that for the whole cohort (AUC 0.65, 0.57 and 0.61, respectively). The optimal cutoff values for discrimination according to the Youden index are presented in Figure 3.

## 4. Discussion

In this retrospective registry of ICU patients with pneumonia undergoing invasive respiratory sampling by BAL, relevant bacteria could be cultivated in 24.4% of samples despite preceding antibiotic therapy in 96%. No correlations of markers of infection, including PCT, CRP and WBC counts, with the microbiological result of BAL were identified in the whole collective of ICU patients. This result remained unchanged when the quantity of cultured bacteria (≥10^4^ CFU/mL) was included in the analysis.

This finding might seem counterintuitive since PCT and CRP levels have been correlated with mortality in pneumonia [13,14,15]. However, an analysis of the literature addressing the diagnosis of HAP and the use of markers of inflammation in this context showed that current guidelines for the management of adults with HAP [3] suggest an estimated area under the ROC of around 0.76 for the diagnosis of HAP with PCT after pooling evidence from trials [15,16,17,18,19,20,21]. Since no scientifically proven PCT-cutoff value could be determined, the overall accuracy was mediocre and bias had to be presumed. A recommendation against incorporating markers of inflammation into decision-making for the diagnosis of pneumonia was made [3]. Similar results have been reported when evaluating CRP or PCT directly from BAL fluid, where no correlation between the level of the inflammatory marker and the diagnosis of VAP or the presence of local or systemic bacteremia could be found [17,22]. Therefore, our findings are in line with already published data that affirm that markers of infection—namely PCT—might be of limited use for the diagnosis of pneumonia in critically ill patients. The high proportion of patients on antibiotic therapy for <24 h at the time of BAL, and the consecutive effect on inflammatory markers, might be attributable for the difference in AUC values between our results and published data.

Considering that we found a better predictive value, especially for PCT, in immunocompromised patients, one might speculate that PCT is of use in these patients. Literature on PCT in immunocompromised patients, in general, is sparse. Some studies demonstrate a good correlation of PCT levels on the first ICU day with sepsis (AUC 0.85) [23], while others report much weaker correlation in immunocompromised pediatric patients [24]. Generally, the usefulness of PCT (and other markers of infection) in immunocompromised patients has been questioned [25,26]. The very high cutoff point for PCT found by the Youden index in our data and the fact that the PCT baseline is elevated in many different diseases, including renal and cardiac dysfunction as well as in immunocompromised patients, make interpretation of PCT levels even more complex [26]. It has been suggested that high CRP values in combination with low PCT values might be indicative of invasive fungal infections [27]; the same might be true for viral infections. Therefore, a more complex approach might be appropriate in pneumonia diagnosis. For this research, blood markers of infection were investigated. Some data suggest that PCT derived from alveolar fluid might be superior for pneumonia detection compared with PCT derived from peripheral blood [28,29] and should be addressed in further trials. It has been suggested that intracellular organisms found in cells in lavage fluid are a specific marker for pneumonia [30], a marker which was not evaluated in our samples. We therefore cannot comment if blood markers of infections correlate with intracellular organisms in lavage fluid.

Importantly, most patients included in this registry were on some form of antibiotic therapy at the time of bronchoscopy. This is clearly based on the fact that BAL were exclusively performed by respiratory specialists during working hours, while patients worsened throughout the whole day, necessitating an empiric antibiotic therapy before BAL was recommended by the Intensivist in charge. Therefore, we cannot exclude that more bacteria might have been cultured if BAL had always been performed in antibiotic-naïve patients. Nevertheless, the fact that even under antibiotic therapy nearly one out of four BAL revealed relevant bacteria and that serum markers of infection could not be used to rule out a positive culture. This might be the rationale for performing a BAL when a microbiological sample is required.

## 5. Limitations

Several limitations have to be considered when interpreting the results of the present registry. Firstly, the retrospective nature of the study imposes a considerable bias on the data. Since only microbiological samples of patients who underwent BAL could be analyzed, the number of pathogens found in patients not undergoing BAL could not be evaluated. In addition, the inclusion of patients over a period of almost ten years entailed the risk of changing management strategies and indication for BAL, with an impact on the statistical correlations. Reasons for not performing BAL include instability of the patient, or the pretest probability for a result of the BAL to influence therapeutic interventions being considered too low by the physicians in charge. Of all patients included in the registry, 93% had been diagnosed with pneumonia and 96% were on antibiotic treatment. Due to the retrospective nature of the study and the sometimes-conflicting documentation of the exact time of BAL in the electronic patient file and bronchoscopy result, we could not differentiate between an antibiotic given right after the BAL and an antibiotic given right before the BAL. We could not retrospectively determine whether a potentially relevant pathogen found in the BAL really caused pneumonia, even if this seemed plausible.

## 6. Conclusions

In this retrospective registry of microbiological cultures derived by bronchoalveolar lavage, a total of 24.4% of all samples showed relevant bacteria. There were no correlations with markers of infection and relevant bacteria. Our data, therefore, might suggest that indication for BAL should not be based on blood markers of infection. In immunocompromised patients, markers of infection might be of some predictive use.

## Figures and Tables

**Figure 1 jcm-10-00486-f001:**
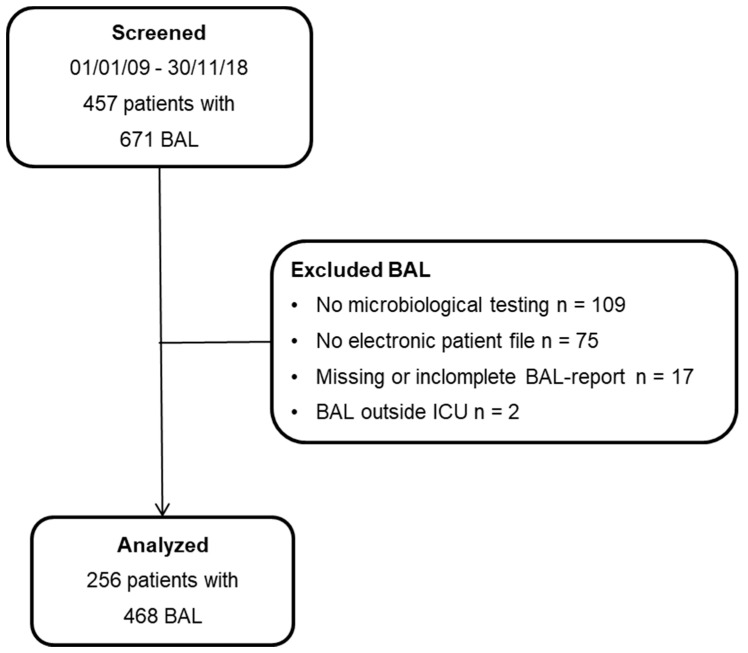
Patients included. Flow chart showing patients undergoing bronchoalveolar lavage (BAL) who were screened and finally included in the registry.

**Figure 2 jcm-10-00486-f002:**
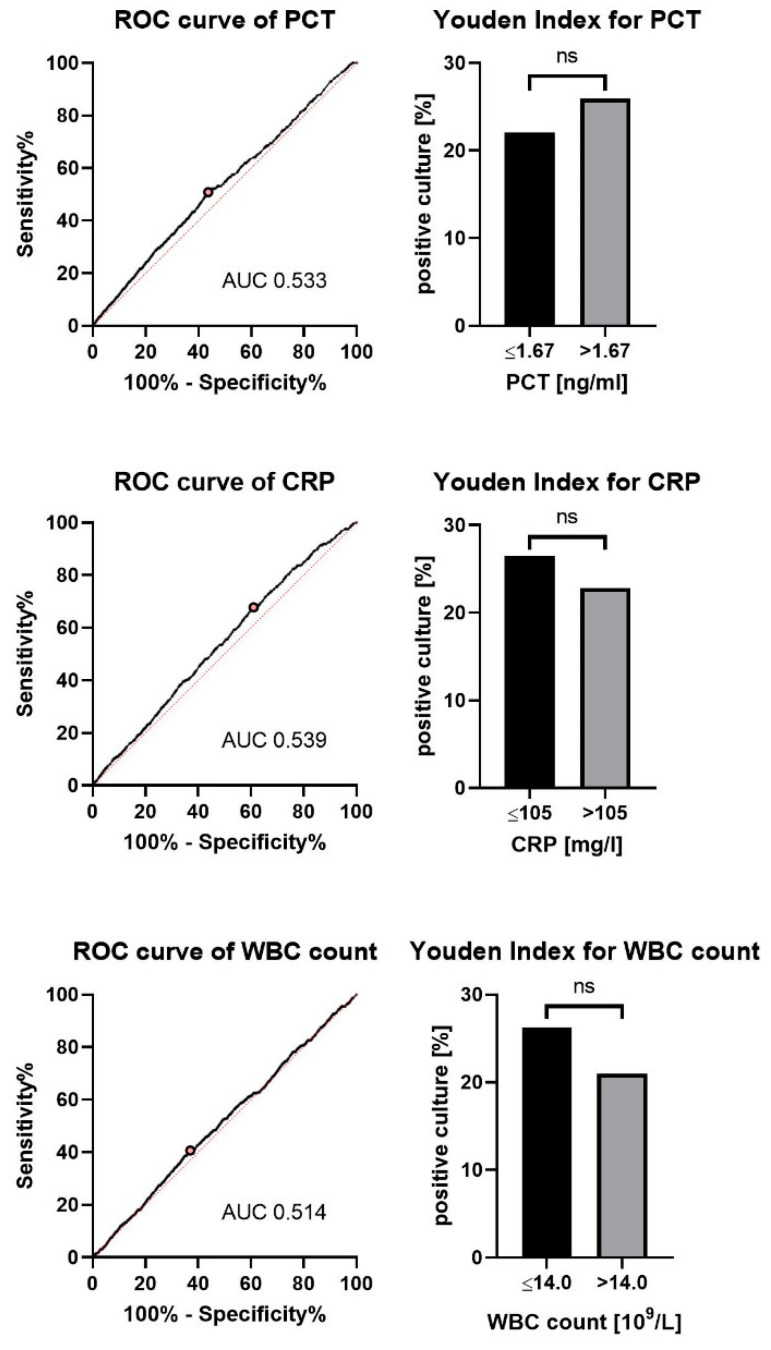
Markers of inflammation for the whole group. ROC curve showing the diagnostic values of PCT and CRP culture positive BAL with relevant bacteria. Best discrimination between the groups according to Youden Index is given for each marker. Abbreviations: BAL = bronchoalveolar lavage, WBC = while blood cell, CRP = C-reactive protein, PCT = procalcitonin.

**Figure 3 jcm-10-00486-f003:**
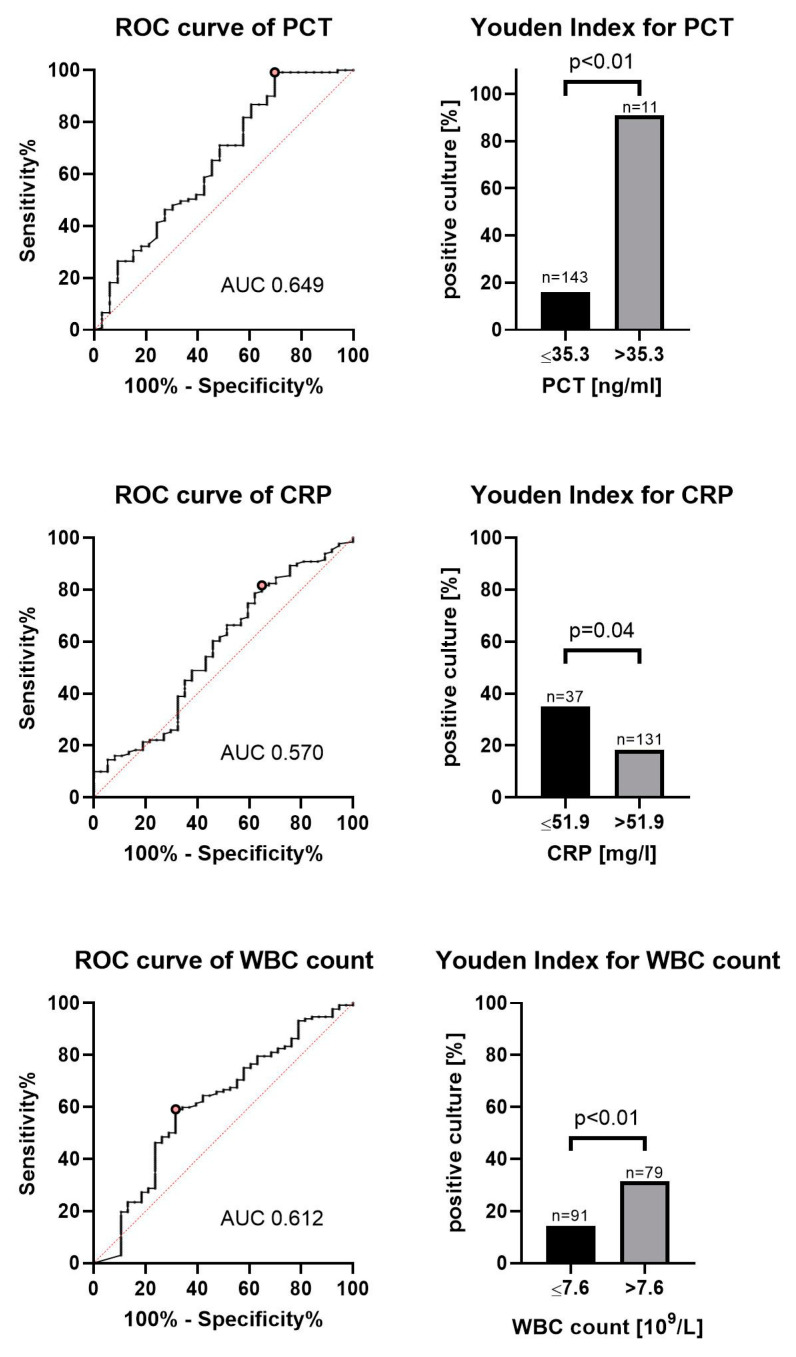
Markers of inflammation in immunocompromised patients. ROC curve showing the diagnostic values of BAL samples positive with relevant bacteria in immunocompromised patients. The best discrimination between the groups according to the Youden Index is given for each marker. Abbreviations: BAL = bronchoalveolar lavage, WBC = while blood cell, CRP = C-reactive protein, PCT = procalcitonin.

**Table 1 jcm-10-00486-t001:** Patient characteristics.

Characteristics	Whole Population	Relevant Bacteria in BAL	No Relevant Bacteria in BAL	*p*-Value
Number of patients	276	62	194	
Number of BAL	468	114	354	
Average number of BAL per patient	1.80	1.80	1.78	1
Male gender	337 (72.0%)	82 (71.9%)	255 (72.0%)	1
ICU-mortality	195 (41.7%)	52 (45.6%)	143 (40.4%)	0.3282
SAPS 2—Score	47 (20–95)	45 (20–79)	48 (24–95)	0.1437
TISS 10—Score	21 (3–40)	21 (9–36)	21 (3–40)	0.9791
Length of ICU stay (days)	17.8 (0.3–84.3)	12.8 (0.7–84.3)	18.2 (0.3–70.8)	0.4452
Time from admission to BAL (days)	3.6 (0–64.1)	2.8 (0–64.1)	3.8 (0–53)	0.6754
Pneumonia diagnosed at time of BAL	435 (92.9%)	110 (96.5%)	325 (91.8%)	0.0963
Immunocompromised	170 (36.3%)	38 (33.3%)	132 (37.3%)	0.5021
Patients on mechanical ventilation	442 (94.4%)	111 (97.4%)	331 (93.5%)	0.1579
Duration of mechanical ventilation (days)	4.6 (0–451)	3 (0–450.6)	5.4 (0–378.8)	0.3233
Patients on corticosteroids at time of BAL	225 (48.1%)	41 (36.0%)	184 (52.0%)	0.0035
Patients on antibiotics at time of BAL	450 (96.2%)	111 (97.4%)	339 (95.8%)	0.5812
No. of antibiotics in used at time of BAL	2 (0–7)	2 (0–7)	2 (0–5)	0.9432
Patients on antimycotic at time of BAL	138 (29.5%)	17 (14.9%)	121 (34.2%)	0.0001
PCT (ng/mL)	1.25 (0–601.2)	1.6 (0–476.9)	1.2 (0–601.2)	0.0724
PCT out of normal range	320 (76.2%)	78 (78.0%)	242 (75.6%)	0.6878
CRP (mg/l)	136.3 (0–563)	124 (4–554)	138 (0–563)	0.7029
CRP out of normal range	451 (96.4%)	110 (98.2%)	342 (98.0%)	1
Median WBC (×10^3^/^mL^)	11.3 (0–190)	11 (0–58.4)	11.3 (0–190)	0.7683
WBC out of normal range	321 (68.6%)	74 (64.9%)	247 (69.8%)	0.3541

Characteristics of all patients included in the analysis. Patients were divided into two categories: one with relevant bacteria in BAL culture and the other without relevant bacteria. Significance was calculated by comparing patients with and without the relevant pathogen. Data are given as the median (range) or number of patients (percent of groups). Abbreviations: BAL = bronchoalveolar lavage, ICU = intensive care unit, SAPS = simplified acute physiology score, TISS = therapeutic intervention scoring system, PCT = procalcitonin, CRP = C-reactive protein, WBC = while blood cell.

**Table 2 jcm-10-00486-t002:** Outcomes of microbiological cultures.

	Whole Population
Number of BAL performed	468 (100%)
Any bacteria cultivated	220 (47.0%)
Relevant bacteria	114 (24.4%)
Any microorganism cultivated *	303 (64.7%)
Relevant microorganism *	126 (26.9%)
No. of microorganisms *	511
No. of relevant microorganism *	167
No. of microorganisms per BAL *	1 (0–5)
No. of relevant microorganisms per BAL *	0 (0–4)
Any fungi cultivated	199 (42.5%)
Relevant fungi	18 (4.9%)

Results of microbiological culture are shown * including bacterial as well as fungal cultures. Abbreviations: BAL = bronchoalveolar lavage.

## Data Availability

The data presented in this study are available on request from the corresponding author.

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
