# Peer review of "Bronchoalveolar Lavage and Blood Markers of Infection in Critically Ill Patients—A Single Center Registry Study"

_jcm, 2021, doi:10.3390/jcm10030486_

Round 1
Reviewer 1 Report
The authors present a retrospective database analysis from a high volume tertiary ICU. They examined the performance of three inflammatory markers in the diagnosis of pneumonia in critical care patients.
The diagnosis of pneumonia was established by BAL and quantitative analysis of the specimens.
The approach and conductance of the study is well described.
Approval by the ethics commission is reported.
The number of patients in the analysis is substantial.
Statistical analysis is explained well and comprehensible as far as I can assess being a clinician.
I advise textual improvement with regard to the indication of diagnostic BAL.
66: “In addition, BAL is strongly advocated in all patients with severe pulmonary failure...”
The question / aim of the study should be formulated more precisely, especially in the abstract.
19: “ It is unclear if diagnostic performance of BAL during working hours is reasonable in ICU patients”
Conclusion and massage for the clinical practice should be more explicit.
It is stated that markers of inflammation (CRP, PCT, WBC) have a poor correlation with results from BAL specimens in suspected pneumonia in critical care patients. This is in line with earlier publications about inflammatory markers in ICU patients in general. (more citations, literature could be included)
I wonder if the mentioned parameters were analysed independently or in combination and correlation with clinical scores as well?
It is an interesting finding that quantitative culture delivered positive findings in 24% of all cases even if 96% of all patients already received antibiotic therapy the time of BAL. Additional information over timing of antibiotic application and BAL would allow better conclusions and interpretation. Can that be retrieved from the database?
Preferably diagnostic procedures (taking specimens) should precede (empiric) antibiotic therapy. In the study population it is reported that 96% of patients already received antibiotic therapy before microbiological cultures by BAL were obtained. This should be scrutinised.
It is explained by the authors that the safety of an invasive diagnostic technique like BAL asks for special skills and training of operators to make it a safe procedure. This restricted the application of BAL in their centre to “office hours”. Furthermore in patients with high ventilatory settings BAL cannot always be performed immediately.
Has it been considered to use the percentage of intracellular organisms (>2% or >5%) in lavage fluid as an additional diagnostic criterion for bacterial infection in patients with delayed BAL after starting antibiotic therapy?
Tracheal aspirates have been taken for culture before commencing antibiotic therapy. What is the level of agreement with later obtained BAL results?
What is the recommendation of the authors after their study with regard to diagnostic BAL and application of inflammatory markers in the diagnostic routine in critical care patients to establish pneumonia in ICU patients?
Author Response
Reviewer 1
Comments and Suggestions for Authors
The authors present a retrospective database analysis from a high volume tertiary ICU. They examined the performance of three inflammatory markers in the diagnosis of pneumonia in critical care patients.
The diagnosis of pneumonia was established by BAL and quantitative analysis of the specimens.The approach and conductance of the study is well described.Approval by the ethics commission is reported.The number of patients in the analysis is substantial.Statistical analysis is explained well and comprehensible as far as I can assess being a clinician.
- I advise textual improvement with regard to the indication of diagnostic BAL. 66: “In addition, BAL is strongly advocated in all patients with severe pulmonary failure...” The question / aim of the study should be formulated more precisely, especially in the abstract. 19: “ It is unclear if diagnostic performance of BAL during working hours is reasonable in ICU patients”
Thank you for pointing this out. We improved the wording regard to the indication of diagnostic BAL. the lines have been adapted and now read
19: It is debated, if blood markers of infection can predict outcome of BAL on a medical ICU.
66: By local policy, a BAL is strongly advocated in all patients with unclear severe pulmonary failure or ARDS and presumed pulmonary infection.
Conclusion and massage for the clinical practice should be more explicit.
We rephrased the conclusion to be more pointed.
Abstract and Conclusion: Our data therefore might suggest that indication for BAL should not be based on blood markers of infection.
- It is stated that markers of inflammation (CRP, PCT, WBC) have a poor correlation with results from BAL specimens in suspected pneumonia in critical care patients. This is in line with earlier publications about inflammatory markers in ICU patients in general. (more citations, literature could be included)
Indeed, our finding is in line with other literature. Three further studies evaluating the usefulness of PCT for the diagnosis of HAP have been included. If reviewer 1 has further studies in mind that should be discussed, we would be happy to include them. Included studies are:
Luyt CE, Combes A, Reynaud C, et al. Usefulness of procalcitonin for the diagnosis of ventilator-associated pneumonia. Intensive Care Med 2008; 34:1434–40.
Liao X, Kang Y. Prognostic value of procalcitonin levels in predicting death for patients with ventilator-associated pneumonia. Intensive Care Med 2010; 36
Zhou CD, Lu ZY, Ren NZ, Zhang GC. Diagnostic value of procalcitonin in ventilator associated pneumonia [in Chinese]. Chin Crit Care Med 2006; 18:370–2.
- I wonder if the mentioned parameters were analysed independently or in combination and correlation with clinical scores as well?
In our analysis we reviewed the mentioned parameters independently from clinical scores. This fact could have been more clearly displayed in the methods section. As displayed, SAPS2 score in our study cohort had no significant correlation regarding a positive BAL result (as displayed in table 1). Our main intention was to find out if there was a direct correlation between the analyzed parameters (WBC, PCR, CRP) and our main endpoint (relevant pathogen detection in BAL culture). The methods were revised:
108: As for this research, outcomes reported were not adjusted for clinical scores or risk factors for pneumonia.
- It is an interesting finding that quantitative culture delivered positive findings in 24% of all cases even if 96% of all patients already received antibiotic therapy the time of BAL. Additional information over timing of antibiotic application and BAL would allow better conclusions and interpretation. Can that be retrieved from the database?
That is a pressing question. When designing our research, we planned on investigating the subgroup of patients undergoing antibiotic therapy and those without antibiotics. However, when we started our study it soon became apparent that many patients that underwent treatment at our ICU were transferred from other hospitals and that data regarding antibiotic treatments was severely lacking in this retrospective analysis. Given that many BAL were also done right after or a few days after ICU admission (see also figure below) we cannot supply this important information.
- Preferably diagnostic procedures (taking specimens) should precede (empiric) antibiotic therapy. In the study population it is reported that 96% of patients already received antibiotic therapy before microbiological cultures by BAL were obtained. This should be scrutinized.
We too were surprised and shocked by that finding. After our analysis of the current practice, the staff on our ICU was reinstructed and programs for antibiotic stewardship were reinforced. We clarified this in the manuscript:
91: Since only final time and date of the microbiological reports were available to us and preliminary results are often communicated directly via telephone, we cannot evaluate if an antibiotic switch was performed as reaction on a preliminary microbiological report or if the switch was an educated guess (empiric) of the physicians in charge.
It is explained by the authors that the safety of an invasive diagnostic technique like BAL asks for special skills and training of operators to make it a safe procedure. This restricted the application of BAL in their centre to “office hours”. Furthermore in patients with high ventilatory settings BAL cannot always be performed immediately.
Indeed.
- Has it been considered to use the percentage of intracellular organisms (>2% or >5%) in lavage fluid as an additional diagnostic criterion for bacterial infection in patients with delayed BAL after starting antibiotic therapy?
As far as our center is considered, the amount of intracellular organisms in lavage fluid is not routinely evaluated. According to our documentation no sample had been tested accordingly. However, as literature shows this parameter to be a good as an early marker for pneumonia in BAL samples, we are considering to incorporate it in future evaluations. The discussion section was edited to feature this consideration.
192: It has been suggested that intracellular organisms found in cells in lavage fluid are a specific marker for pneumonia [30], a marker which was not evaluated in our samples. We therefore cannot comment if blood markers of infections correlate with intracellular organisms in lavage fluid.
- Tracheal aspirates have been taken for culture before commencing antibiotic therapy. What is the level of agreement with later obtained BAL results?
Thank you for this question. When planning this research, we also wanted to address this important question. Unfortunately, only a subgroup of patients had both, Tracheal aspirates and BAL performed within a short period of time. Since patient numbers are limited in this subgroup, we cannot report reliable data.
- What is the recommendation of the authors after their study with regard to diagnostic BAL and application of inflammatory markers in the diagnostic routine in critical care patients to establish pneumonia in ICU patients?
In our research, there was no correlation between significant pathogens found in BAL and blood markers for infection. Importantly, only critically ill patients were included, which might have various reasons for elevated serum markers of infections like coinfections. Therefore, in this specific subgroup of patients, we clinically diagnose pneumonia and initiate antibiotic therapy disregarding serum markers of infections.
Reviewer 2 Report
Thank you for asking me to review this manuscript which reports the results from diagnostic investigation of hospital-acquired pneumonia by bronchoalveolar lavage in a single centre ICU. The authors report the results of a retrospective review of patients undergoing bronchoscopic investigation for pneumonia, and examined the ability of peripheral blood markers of infection to predict detection of clinically relevant pathogens on culture.
- Although the others indicate the operators of the bronchoscopy service, it would be helpful to know if they had a standard operating procedure for lavage (choice of segment to be lavaged, volumes of fluid instilled, whether initial aliquot was discarded, whether more than one segment could be lavaged).
- In table 1 it is noted the % of patients who had pneumonia diagnosed at time of lavage – presumably this was by clinical and radiological criteria? Could the authors clarify the criteria they used to diagnose pneumoniae and whether these were based on established criteria such as those from the European CDC?
- It would be helpful to have a summary of the microbiological approach taken to the samples, specifically what tests were conducted, how were bacteria identified (e.g. MALD-TOF), where any PCR or antigen (e.g. glactomannan or Beta-d-glucan) tests conducted as standard?
- The authors note in table 2 that fungi were cultured, including 5% clinically relevant. For completeness the total fungi cultured (presumably mostly candida) should be included in supplemental table 2 – and the clinically relevant fungi should be included in supplemental table 1
- Did the ROC curves generated include the 18 clinically relevant fungi, and if so do the authors think this may impact on the performance of PCT?
- Although the authors report the percentage of patients on antibiotics at time of lavage, negative cultures are most frequent following a recent change in antibiotics – do the authors have data on the duration of antibiotics prior to lavage rather than just the percentage of patients who had received antibiotics?
- There is some evidence of compartmentalisation of the inflammatory response in pneumonia, with studies of alveolar inflammatory markers having better diagnostic performance than peripheral blood markers (e.g Conway Morris A, et al Thorax. 2010;65(3):201-207 and Millo, J.L., et al Intensive Care Med2004 30, 68–74. ) The authors should discuss their results in the context of this proposed compartmentalisation, and its implications for the use of lavage in diagnosing pneumonia.
Minor comments
Table 1 – it appears that the lines for median PCT, CRP and WCC counts and % with abnormal results have become mixed up.
Figure 3 appears to be an error, showing part of the recruitment diagram and not the ROC curves for immunocompromised patients?
Author Response
Reviewer 2
Comments and Suggestions for Authors
Thank you for asking me to review this manuscript which reports the results from diagnostic investigation of hospital-acquired pneumonia by bronchoalveolar lavage in a single centre ICU. The authors report the results of a retrospective review of patients undergoing bronchoscopic investigation for pneumonia, and examined the ability of peripheral blood markers of infection to predict detection of clinically relevant pathogens on culture.
- Although the others indicate the operators of the bronchoscopy service, it would be helpful to know if they had a standard operating procedure for lavage (choice of segment to be lavaged, volumes of fluid instilled, whether initial aliquot was discarded, whether more than one segment could be lavaged).
Thank you very much for pointing this out. We expanded the manuscript by a paragraph named “Methods of BAL”
Methods BAL: BAL was performed as reported previously [11] and carried out in the radiologically most affected lung lobe. In cases of diffuse infiltrates or involvement of multiple lobes, the middle lobe or the lingual were preferred for BAL. Bronchoscopes with a diameter of 8 mm were used to obtain a standardized wedge position. For exclusive microbiological sampling 100 ml of sterile saline (0.9% NaCL) were instilled in 20 ml aliquots. After each instillation, the fluid was gently suctioned. If immunological or ad-ditional analysis was warranted BAL was performed with up to 300 ml of sterile saline. The BAL aliquots (including the initial aliquot) were pooled and collected in a sterile jar and immediately transported to the laboratory for further analysis.
- In table 1 it is noted the % of patients who had pneumonia diagnosed at time of lavage – presumably this was by clinical and radiological criteria? Could the authors clarify the criteria they used to diagnose pneumoniae and whether these were based on established criteria such as those from the European CDC?
Thank you for this remark. We should have been clearer with regards to our definition of pneumonia. Concerning the percentage of patients with pneumonia we had to rely on documentation of the physicians in charge of treatment of the analyzed patients. If the presence of “pneumonia” was described in the physician’s letter regarding the ICU stay that patient was considered to have had pneumonia regarding our analysis. The criteria on which a pneumonia is diagnosed at out center are those specified by the German guideline for hospital acquired.
We edited the methods section accordingly:
78: Pneumonia was considered to be present when documented by the physician in charge as diagnose in the electronic patients files. The diagnosis pneumonia was based on the current German guidelines for HAP [11, 12].
- It would be helpful to have a summary of the microbiological approach taken to the samples, specifically what tests were conducted, how were bacteria identified (e.g. MALD-TOF), where any PCR or antigen (e.g. glactomannan or Beta-d-glucan) tests conducted as standard?
Thank you for pointing this out. Specifically, the lavage fluid samples were, after collection; send off to be processed by the institute for microbiology. The samples were then cultured using different standard microbiological agar media such as Columbia blood agar, chocolate agar, MacConkey agar, yeast cysteine blood (anaerobic) agar, GVPC agar (Legionella) and Sabouraud dextrose agar (fungal diagnostic). The identification of pathogens was done by using MALDI-TOF. For the differentiation of Streptococcus pneumoniae from other Streptococcus spp. the optochin susceptibility test was further utilized.PCR and antigen test were not conducted as a standard procedure. However, 324 out of 468 (69%) BAL were further analyzed by PCR or antigen tests. 100 of those returned a relevant pathogen (57% viral, 34% fungal, 9% bacterial). PCR was mostly conducted in to help the diagnose of PJP or to find viral pathogens in immunosuppressed patients. Antigen tests were mostly used to detect Aspergillus.
The methods section was further edited to accompany the procedure of identifying bacteria by the institute for microbiology.
After the lavage fluid had been recovered, it was processed by the microbiological institute of the University of Freiburg. Routine diagnostics were performed for each sample, which included the creation of microbiological cultures on different standard media (blood agar, chocolate agar, Mac-Conkey agar, yeast cysteine blood agar, GVPC agar and Sabouraud dextrose agar). The Identification of species was done by using MALDI-TOF. In case of Streptococcus pneumoniae differentiation from other Streptococcus spp. was achieved by using the optochin susceptibility test. The results of the microbiological diagnostics are documented in the electronic patient files.
- The authors note in table 2 that fungi were cultured, including 5% clinically relevant. For completeness the total fungi cultured (presumably mostly candida) should be included in supplemental table 2 – and the clinically relevant fungi should be included in supplemental table 1
That assumption is correct. Most detections of fungi indeed belonged to Candida spp.. Relevant fungi consisted mostly of detections of Aspergillus in culture. Data from PCR and Antigen tests was not included in our publication as it had not been done routinely. To provide a better overview of the results we now included all relevant and non-relevant fungi in the respective supplemental tables.
- Did the ROC curves generated include the 18 clinically relevant fungi, and if so do the authors think this may impact on the performance of PCT?
Indeed, the 18 patients with clinically relevant fungi were included in this registry. When excluding these 18 patients, similar results were obtained (see ROC curves below). This might be explained by the fact, that from this 18 patients, 6 were diagnosed with a significant bacteria as well. This information is now given in the SEM
- Although the authors report the percentage of patients on antibiotics at time of lavage, negative cultures are most frequent following a recent change in antibiotics – do the authors have data on the duration of antibiotics prior to lavage rather than just the percentage of patients who had received antibiotics?
This is an important remark. Initially we planned on assessing antibiotic therapies with regards to the duration and influence on BAL results. Unfortunately, most patients had already been given an antibiotic in the 24h leading up to the BAL. Furthermore, information was scarce regarding the length of the antibiotic therapy before BAL. This was mostly due to the fact that many patients had been transferred from other hospitals or wards to our ICU and that external information wasn’t able to be retrieved. Furthermore, most BAL were done shortly after admission (see figure below). We therefore decided not to include data on the duration of antibiotic therapies in our analysis as it was considered to be insufficient. Therefore, the requested data cannot be given.
- There is some evidence of compartmentalisation of the inflammatory response in pneumonia, with studies of alveolar inflammatory markers having better diagnostic performance than peripheral blood markers (e.g Conway Morris A, et al Thorax. 2010;65(3):201-207 and Millo, J.L., et al Intensive Care Med2004 30, 68–74. ) The authors should discuss their results in the context of this proposed compartmentalisation, and its implications for the use of lavage in diagnosing pneumonia.
We have to agree to reviewer 1 that data exists suggesting a better performance of alveolar PCT compared to bloodborne PCT. This important fact and both citations are now discussed in the manuscript. We included the following sentence in the discussion.
For this research, blood markers of infection were investigated. Some data suggests that PCT derived from alveolar fluid might be superior for pneumonia detection compared to PCT derived from peripheral blood [26, 27] and should be addressed in further trials.
Minor comments
- Table 1 – it appears that the lines for median PCT, CRP and WCC counts and % with abnormal results have become mixed up.
Indeed. The values for the last two columns were switched by accident. This has now been fixed. Thank you very much
- Figure 3 appears to be an error, showing part of the recruitment diagram and not the ROC curves for immunocompromised patients?
Thank you very much for this hint. Indeed, the wrong figure was uploaded initially. Now, the real figure 3 is supplied.

Round 2
Reviewer 2 Report
no further comments